# Cognitive-bias modification intervention to improve physical activity in patients following a rehabilitation programme: protocol for the randomised controlled IMPACT trial

Boris Cheval [1,2] Axel Finckh,[3] Silvio Maltagliati,[4] Layan Fessler,[4] Stéphane Cullati,[5,6] David Sander,[1,2] Malte Friese,[7] Reinout W Wiers,[8] Matthieu P Boisgontier,[9,10] Delphine S Courvoisier,[11] Christophe Luthy[12]

For numbered affiliations see end of article.

**Correspondence to**
Dr Boris Cheval;
boris.cheval@unige.ch

## ABSTRACT

**Introduction** Being physically active is associated with a wide range of health benefits in patients. However, many patients do not engage in the recommended levels of physical activity (PA). To date, interventions promoting PA in patients mainly rely on providing knowledge about the benefits associated with PA to develop their motivation to be active. Yet, these interventions focusing on changing patients' conscious goals have proven to be rather ineffective in changing behaviours. Recent research on automatic factors (eg, automatic approach tendencies) may provide additional targets for interventions. However, the implementation and evaluation of intervention designed to change these automatic bases of PA are rare. Consequently, little is known about whether and how interventions that target automatically activated processes towards PA can be effective in changing PA behaviours. The Improving Physical Activity (IMPACT) trial proposes to fill this knowledge gap by investigating the effect of a cognitive-bias modification intervention aiming to modify the automatic approach towards exercise-related stimuli on PA among patients.

**Methods and analysis** The IMPACT trial is a single-centre, placebo (sham controlled), triple-blinded, phase 3 randomised controlled trial that will recruit 308 patients enrolled in a rehabilitation programme in the Division of General Medical Rehabilitation at the University Hospital of Geneva (Switzerland) and intends to follow up them for up to 1 year after intervention. Immediately after starting a rehabilitation programme, patients will be randomised (1:1 ratio) to receive either the cognitive-bias modification intervention consisting of a 12-session training programme performed over 3 weeks or a control condition (placebo). The cognitive-bias modification intervention aims to improve PA levels through a change in automatic approach tendencies towards PA and sedentary behaviours. The primary outcome is the sum of accelerometer-based time spent in light-intensity, moderate-intensity and vigorous-intensity PA over 1 week after the cognitive-bias modification intervention (in minutes per week). Secondary outcomes are related to changes in (1) automatic approach tendencies and self-reported motivation to be active, (2) physical health and (3) mental health. Sedentary behaviours and self-reported PA will also be examined. The main time point of the analysis will be the week after the end of the intervention. These outcomes will also be assessed during the rehabilitation programme, as well as 1, 3, 6 and 12 months after the intervention for secondary analyses.

**Ethics and dissemination** The study will be conducted in accordance with the Declaration of Helsinki. This trial was approved by the Ethics Committee of Geneva Canton, Switzerland (reference number: CCER2019-02257). All participants will give an informed consent to participate in the study. Results will be published in relevant scientific journals and be disseminated in international conferences.

**Trial registration details** The clinical trial was registered at the German clinical trials register (reference number: DRKS00023617); Pre-results.

## Strengths and limitations of this study

► The randomised controlled Improving Physical Activity trial will test the effects of an intervention based on cognitive-bias modification to improve physical activity among patients following a rehabilitation programme.
► Physical activity, sedentary behaviours, physical health and mental health will be measured at multiple time points over 1 year.
► The findings from this well-powered study will provide evidence-based recommendations for clinical interventions aiming to promote physical activity among patients in rehabilitation.
► The reliance of a single-centre trial and the selection bias due to lost to follow-up and the volunteer participation are keys limitations that may reduce our ability to generalise the results to other populations.

## INTRODUCTION

The health benefits of physical activity (PA) are well established and extensive. PA can

reduce rates of cardiovascular diseases,[1] cancers,[2] hypertension,[3] diabetes,[4] obesity,[5] depression[6] and all-cause mortality,[7] even more effectively than medication.[8] PA is safe and beneficial for almost everyone, while the risk of harm from moderate PA is small.[8 9] A recent systematic review and meta-analysis suggests that any PA, irrespective of the intensity, is beneficial for health.[7] In patients suffering from chronic diseases, increased PA is associated with reduced hospital admissions, decrease in pain, greater quality of life and mental health, and improvement in physical function.[8 10–13] These myriads of benefits even led the Academy of Medical Sciences to consider PA as a miracle cure.[14] Nevertheless, patients, similarly to the general population, remain largely physically inactive.[15–17]

Healthcare professionals are uniquely placed to promote PA among patients. Today, interventions aiming to enhance PA in patients largely relies on providing rational information about the benefits associated with PA. For example, a practical guide to help clinicians discussing about PA within a consultation has been recently proposed.[8] In this guide, clinicians are encouraged to rationally address patients' concerns about PA, to explain that there are more benefits to become active than to remain sedentary, to set an achievable goal, to identify barriers to be overcome, and finally, to set a plan. This type of intervention guide is grounded in the dominant social-cognitive theories,[18] which contend that goals are proximal determinants of behaviours.[19 20] From these perspectives, changing patients' conscious goals should lead to substantial changes in their behaviours.[21 22] While these types interventions have proven to be effective to change PA behaviours to some extent,[23] meta-analyses also indicate that these approaches are more effective in changing intentions than in changing actual behaviour.[24] Thus, developing additional interventions targeting alternative mechanisms is needed.

Recent research focusing on automatic mechanisms may provide additional targets for interventions.[25–29] For example, studies showed that in physically active individuals stimuli associated with PA attract attention,[30–33] trigger positive affective reactions[34–37] and activate approach tendencies towards PA.[38–41] These automatically activated processes are thought to facilitate the translation of conscious goals into actual PA behaviours. Importantly, these automatic reactions predict PA behaviours above and beyond self-reported measures, such as the intention to be physically active,[39] and are stronger predictors of spontaneous and unplanned actions that often consist of light-intensity PAs.[42] As such, from this perspective, physical inactivity is thought to also result from an imbalance between a strong motivation to be physically active, but weak automatic approach tendencies towards PA. Crucially, this imbalance between automatic and reflective processes may be particularly pronounced in patients, whose automatic reactions towards PA may be negatively biased by the fear, pain and discomfort felt during some exercises.[43] Thus, in comparison with the general population, patients may demonstrate more negative automatic

reactions towards PA, including, for example, stronger negative affective reactions and weaker approach tendencies towards PA. One practical implication of these findings is that interventions designed to promote PA in patients might particularly benefit from directly targeting automatically activated processes towards PA.

What kinds of interventions can target automatically activated processes? New types of interventions have been developed to directly target these automatic reactions towards a given health behaviour.[44 45] For example, in alcohol addiction, studies have used a cognitive-bias modification (CBM) intervention aimed at retraining automatic approach reactions towards alcohol using a computerised task.[46] In a CBM intervention, patients were repeatedly asked to push a joystick when exposed to alcohol-related pictures, simulating an avoidance movement. Specifically, in this computerised-based task, participants were asked to push or pull a joystick in response to the format of the pictures. For example, they were instructed to make a pushing movement when the picture presented on the screen was in the landscape format (ie, avoidance), and to make a pulling movement when the picture was in the portrait format (ie, approach). To ensure congruence with the participant's actions on the joystick, the picture became smaller when the participant pushed the joystick, and it became larger when the participant pulled the joystick. Participants received training in which they had to push the joystick away in response to pictures of alcohol (ie, all alcohol pictures were presented in the push format) and to pull the joystick towards them in response to non-alcohol pictures (ie, all non-alcohol pictures were presented in the pull format). Three large studies conducted in patients showed that adding a CBM intervention to a regular cognitive-behaviour treatment yielded a beneficial effect on the relapse rates 1 year after treatment discharge, with a reduction of 9%,[47] 13%,[46] and 12%,[48] which could be attributed to changes in approach tendencies.[47 49] These interventions have also proven to be useful in impacting cigarette smoking,[50] social anxiety[51] or eating behaviours.[52–54] Yet, it should be noted, the clinical effectiveness of CBM interventions has been criticised,[55 56] especially for anxiety and depression-related outcomes.[57–60]

To the best of our knowledge, however, only a handful set of studies have been conducted to target automatic processes towards PA.[61–64] Crucially, only one study has been conducted to examine the effect of a brief CBM intervention targeting approach-avoidance tendencies on an exercise task in a sample of healthy young adults.[64] Specifically, using a manikin task,[42 65] a variant of the approach-avoidance joystick task, participants were explicitly trained to repeatedly approach a manikin towards pictures depicting PA and to avoid pictures depicting sedentary behaviours, by pressing keys on the keyboard. Results revealed that participants spent more time exercising during a laboratory exercise task of moderate intensity (ie, doing squat), in comparison with control groups either trained to approach stimuli depicting

sedentary behaviours and avoid stimuli depicting PA (ie, reverse contingencies) or to approach and avoid stimuli depicting PA and sedentary behaviours equally often (sham controlled). These findings suggest that a single and brief CBM session targeting automatic approach tendencies towards PA and sedentary behaviours can have beneficial effect on laboratory-based PA behaviours. However, this study has at least two important limitations. First, it is unclear if and to what extent the PA behaviour performed in the laboratory extends to behaviours performed in everyday life, thereby preventing the possibility to determine whether CMB manipulations can be effective in changing daily-life behaviours. Second, the study was conducted on a sample of rather physically active college students. As such the potential beneficial effect of adding a CBM intervention to a regular treatment in patients, a population which may particularly benefit from such manipulation, remains unknown.

## Objectives

In sum, while recent research highlights the importance of targeting automatically activated processes related to PA, the effectiveness of interventions designed to change these presumed automatic bases of PA behaviours has been largely overlooked. Consequently, little is known about whether and how interventions that target automatically activated processes towards PA can be effective in changing behaviours. The primary objective of the Improving Physical Activity (IMPACT) trial is to investigate the effectiveness of a CBM intervention targeting automatic approach tendencies towards exercise-related stimuli on PA patients in a rehabilitation programme. This trial will be performed using a placebo, triple-blinded, phase 3 randomised controlled trial. The secondary objectives are to evaluate the effect of this CBM intervention on changes in (1) automatic approach tendencies and self-reported motivation to be active, (2) physical health and (3) mental health. We hypothesise that the CBM intervention will be associated with higher levels of PA (preintervention vs 1-week postintervention) (H1). Moreover, we hypothesise that the CBM intervention will increase automatic approach tendencies towards PA (H2a), but will decrease automatic approach tendencies towards sedentary behaviours (H2b). Finally, we predict that the CBM intervention will improve patients' physical and mental health (H3). All these hypotheses will also be tested during the rehabilitation programme as well as 1, 3, 6 and 12 months after the intervention (secondary analyses).

## METHODS AND ANALYSIS
### Study design

The IMPACT trial is a single-centre, placebo (sham controlled), triple-blinded, phase 3 randomised controlled trial. The trial will start (First-Participant-In) January 2022 in the ward 3DK of the Division of General Medical Rehabilitation (University Hospitals of Geneva;

---

> **Box 1  Inclusion and exclusion criteria**
>
> **Inclusion criteria**
> ► Patients treated in the ward 3DK of the Division of General Medical Rehabilitation.
> ► Aged 18 years or older.
> ► Can comply with the study protocol.
> ► Able to provide a written consent of participation in the trial.
>
> **Exclusion criteria**
> ► Contraindication to physical activity in the view of the health status.

---

Switzerland) and will finish (Last-Participant-Out) in January 2024. The ward 3DK admits and manages patients for treatments or diagnostics evaluations, especially after being in acute care for several reasons, such as serious infections, cancer, heart failure or postsurgery follow-up treatments. This ward offers multidisciplinary treatment in rehabilitation (eg, physiotherapists, occupational therapists, nutritionists) and does not focus on improving PA engagement. In other words, within the usual care, there is not any content specifically devoted to improve patients' PA level. Eligible patients will be randomly assigned to either the CBM intervention or the active control condition (placebo) in a 1:1 ratio. The current study follows the Standard Protocol Items: Recommendations for Interventional Trials statement.[66]

### Eligibility criteria

The eligibility criteria are listed in box 1. Participants fulfilling all the inclusion criteria are eligible for the study. The presence of the exclusion criterion will lead to the exclusion of the participant.

### Decision to include/exclude a participant

The decision to include/exclude a participant from this study will be jointly decided by the chief medical officer and the research assistant.

### Participant screening, recruitment and consent

All patients starting rehabilitation programme in the ward 3DK of the Division of General Medical Rehabilitation, University Hospital of Geneva, Switzerland (from January 2022 to January 2024) will be approached during the first consultation with the chief medical officer and will receive an information sheet explaining the main objective of the IMPACT trial. The investigators will explain to each participant the nature of the study, its purpose, the procedures involved, the expected duration, the potential risks and benefits and any discomfort it may entail. Each participant will be informed that the participation in the study is voluntary and that he or she may withdraw from the study at any time and that withdrawal of consent will not affect his or her subsequent medical assistance and treatment. The participant will be informed that his or her medical records may be examined by authorised individuals other than their treating physician. All participants will be provided a participant information sheet

**Table 1** Overview of the baseline screening measures

| Measures | Assessment method |
|---|---|
| **Inclusion criteria** | |
| Patients treated in ward 3DK of the Division of General Medical Rehabilitation | During the first meeting with the research assistant |
| ≥18 years of age | |
| Can comply with study protocol | |
| Able to provide a written consent | |
| **Exclusion criterion** | |
| Contraindication to PA in the view of the health status | During the first meeting with the research assistant |
| **Additional baseline screening assessment** | |
| Medical evaluation (questionnaires and objective tests) | Patients' diseases and treatment characteristics (medical burden, comorbidity, body mass index, mobility test, functional independence, health-related quality of life) |
| Sociodemographic characteristics | Questionnaires (age, sex, height, weight) |
| Usual level of PA | Saltin-Grimby PA Level Scale.[81] |
| Personality | Ten-Item Personality Inventory.[82] |
| Expectations for improvement | A questionnaire measuring patients' thoughts about the effects of the intervention (three items: 'to what extent do you think that your physical activity behaviors will improve as a result of training on the computerized task?'; 'to what extent do you think that your mental health will improve as a result of training on the computerized task?'; 'to what extent do you think that your physical health will improve as a result of training on the computerized task?').[67] |
| Self-reported motivation to change | Questionnaire measuring patients' motivation to change their condition (two items: 'how motivated are you to change your health condition?'; 'to what extent do you really want to change your health condition?'), to avoid a new treatment (two items: 'how motivated are you to avoid a new treatment because your health condition?'; 'to what extent do you really want to avoid taking a new medication because of your health condition?', and to engage in more PA in the future(two items: 'I intend to carry out more physical activity in the next future'; I am determine to carry out more physical activity in the next future').[66] |
| Self-reported ability to implement daily-life PA | Questionnaire measuring patients' self-reported ability to adopt regular PA in their daily life. Self-reported function in instrumental activities of daily life (ADL; seven items), in activities of daily living (ADL; seven items), and in mobility (three items).[83] |

PA, physical activity.

and a consent form describing the study and providing sufficient information for participants to make an informed decision about their participation in the study (see online supplemental material 1 the patient consent form). Participants will have time to carefully read the documents and can give their responses up to 24 hours after having received the documents. The formal consent of a participant, using the approved consent form, will be obtained before the participant is submitted to any study procedure. Participants will then complete a first questionnaire assessing the exclusion and inclusion criteria, as well as other screening measures. All the questionnaires will be assessed electronically using REDCap software. Finally, patients' expectations regarding the effects of the intervention will be assessed.[67] Table 1 provides an overview of all the baseline screening measures available. The study patient flow chart is provided in figure 1.

### Sample size

For power calculation, our intervention implements a between-subject design and random-effects statistical models (ie, t-tests). The power calculation is based on the primary outcome (ie, accelerometer-based time spent in light-intensity, moderate-intensity and vigorous-intensity PA over 1 week after the CBM intervention (in minutes per week)). Based on estimates of the effect size of interventions targeting automatic approach tendencies (ie, Cohen's d=0.41; eg, a difference of ~30 min per week between the intervention and the control group for a pooled SE of ~75 min per week),[68 69] a sample size

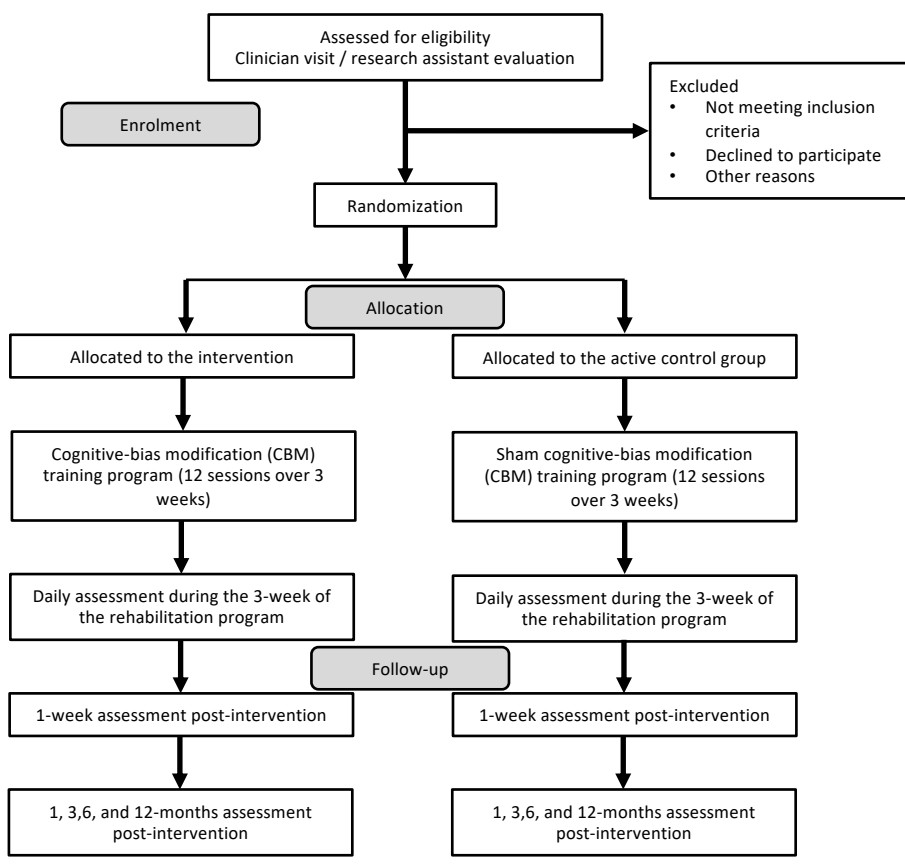

Figure 1 Flow chart note. The daily assessment refers to the measure of PA behaviours that will be continuously assessed during the rehabilitation period. The secondary outcomes will be assessed on a weekly basis. PA, physical activity.

calculation indicates that a minimum of 252 patients (126 per arm) would be needed to demonstrate efficacy of the intervention on the device-based PA during the week following the intervention, with a probability of committing a type I error <5% and a probability of committing a type II error <10%. We expect a loss to follow-up of 10%–20% at 1 week after the intervention, and a loss of 30%–40% over 1 year. Thus, a minimum of 352 patients will be recruited. Of note, with this sample size, an alpha of 0.05 and a power of 0.90, the smallest effect size we could detect is d=0.35. Finally, using the expected effect size (ie, d=0.41), an alpha of 0.05 and a sample size of 352 patients, we obtain a power of 0.97.

## Feasibility

The ward 3DK of the Division of General Medical Rehabilitation has 24 beds and treats on average 40 patients per month. Based on the chief medical officer's experiences and a first presentation of the study to the patients treated in this unit, we expect that three patients out of five will not agree (for various reasons) to participate in the study, thereby leading to a total of about

24 participants recruited per month. Consequently, we should be able to collect the target sample size in aapproximately 15–17 months. The average duration of participants hospitalisation in the ward 3DK is about 3 weeks. As such, though this duration can vary between patients (ie, some patients only stay a few days), this duration allows for the implementation of the whole intervention (ie, 12-session training programme performed over 3 weeks). Of note, participants who will not complete all the training sessions will still be included in the analysis. Sensibility analyses will be conducted to examine whether the number of completed sessions influence the effects of the intervention. To accelerate and facilitate knowledge dissemination, all articles will be preprinted, and data and code shared on public repositories.

### Patients adherence to the impact trial

Patients adherence to the training programme (ie, if the planned training session is completed or not, and why in case of no completion) and to the other measures are documented in an electronic case report form (eCRF) powered by REDCap.[70] To promote patient retention and complete follow-up (ie, 1, 3, 6 and 12 months after the end of the intervention), participants will be contacted by phone by a research assistant 2 weeks before the follow-up measurement. If they do not answer, they will receive up to two additional phone calls this week. If they do not answer, this procedure will be repeated the following week. If they still do not answer, this time of measurement will be considered as missing. Patients with missing data at a given wave will be contacted for the following waves through the above-mentioned procedure.

Patients who did not answer a given time of measurement, will still be contacted to participate in the following time points.

### Interventions

All newly admitted patients will attend a meeting organised in the unit. The objective of this meeting will be to present and illustrate the health benefits of PA. Consistent with the recent practical guide to help healthcare professionals promoting PA to patients,[8] research assistants will follow the 'Ask-Assess-Advise' structure for discussing PA behavioural change in the consultation. Patients will also receive a watch tracking (ie, polar) during the rehabilitation period and giving personalised feedback on their PA and sedentary behaviours. This procedure aims at increasing their self-reported motivation to be active, thereby allowing to examine the additional effects of the CBM intervention.

Intervention group: Training programme of 12 sessions over 3 weeks (ie, 4 sessions by week on average) using an adapted version of the Visual-Approach/Avoidance-by-the-Self Task,[71] a task that have shown to produce large and replicable effects, compared with the manikin task. Specifically, patients will be asked to react to the format (ie, portrait vs landscape format) of the pictures depicting PA and of sedentary behaviours by pressing

twice the 'move forward' or 'move backward' key press to approach or avoid the pictures, respectively. Participants will be instructed to approach the picture when it appears in a portrait format, and to avoid it when the picture appears in a landscape format (the rule will be counterbalanced between participants). Of note, unlike the previous study that relied on an explicit instruction task (ie, participants were asked to respond to the content of the pictures),[64] the current study uses an irrelevant feature task (ie, participants were asked to respond to the format of the pictures). This irrelevant feature task allows a training without explicit instruction. Congruent with the patient's approach or avoidance response, the whole visual environment will zoom in on the picture to simulate an approach movement and zoom out to simulate an avoidance movement. A change by 10% after each key press will be used to give the impression to walk forward or backward as a consequence of the responses. Participants in the intervention group will receive training in which 90% of pictures depicting PA will be presented in the approach format (and 10% in the avoidance format), and 90% pictures depicting sedentary behaviours will be presented in the avoidance format (and 10% in the approach format). This 90/10 split aims to increase the patients blinding to the condition in which they will be assigned. Each training session will consist of 144 trials for a total duration of approximately 10 min. At the first session and at the beginning of each week, the training session will be preceded by 96 assessment trials in which the contingency of approaching or avoiding PA or sedentary behaviours will be 50%. Assessment trials will allow to measure patients' automatic approach-avoidance tendencies towards PA and sedentary behaviours (see figure 2).

Comparator group: Patients in the comparator group (placebo; sham controlled) will not be trained to approach PA and to avoid sedentary behaviours. Specifically, the retraining sessions will also consist of 144 trials, but the task will require an equal number of approach and avoidance responses to both stimuli depicting PA and sedentary behaviours (see figure 2). The use of a placebo was chosen to ensure that the potential effects of the experimental condition will be attributable to the content of the training programme (ie, learn to systematically approach PA-related stimuli and avoid sedentary behaviours-related stimuli) rather than because of a simple exposition effect (ie, the fact to be exposed longer to contents related to PA and sedentary behaviours).

Stimuli: Stimuli representing PA and sedentary behaviours will be created using the Unity software . A set of 195 pictures including 14 avatars (50% women) in either active (walking and running) and inactive posture (sit on a cubicle) will be tested in a pilot study to identify the 48 pictures the most associated with 'movement and physically active behaviours' and the 48 pictures the most associated with 'rest and physically inactive behaviours' using two Visual Analogue Scale (VAS 1; 'please indicate how this image is, in your opinion, associated with a behaviour that requires: 0=no physical exertion at all,

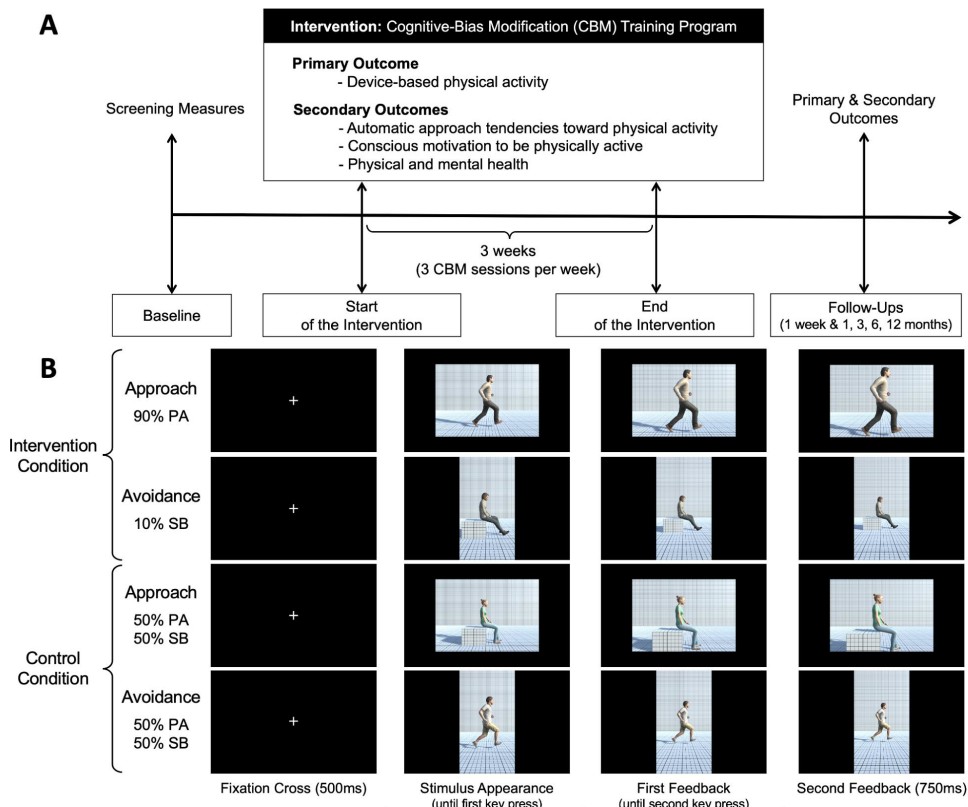

**Figure 2** Study design and of cognitive-bias modification (CBM) task. Note: (A) Study design. (B). Illustration of the CBM task. In the CBM task, participants are asked to approach or avoid the picture appearing on the screen depending on its format (ie, portrait vs landscape format, counterbalanced across participants). Participants are asked to approach the picture in the approach conditions and to avoid the picture in the avoidance conditions. In the intervention condition, 90% of the pictures depicting physical activity are presented in the approach format (10% in avoidance format), and 90% of the pictures depicting sedentary behaviours are presented in the avoidance format (10% in approach format). In the control condition, the pictures depicting physical activity and sedentary behaviours are equally distributed across formats (ie, 50%–50%). PA, physical activity.

100=a lot of physical exertion'; VAS 2; 'please indicate how closely this image is associated with: 0=resting, sedentary behaviour, 100=movement, very active behaviour'). The credibility of the pictures will also be tested ('how realistic do you think this person's behavior is? Realistic meaning that the images may resemble to a real-life behavior'; on a VAS from 0=behaviour not at all realistic; 100=behaviour very realistic) and for agreeableness ('how pleasant/sympathetic do you find the person in this image? For example, would you like to talk to her/him'; from 0=very unpleasant/antipathetic, 100=very pleasant/sympathetic). The aim of this pilot study was twofold. First, to ensure that the selected pictures reflect the concept of interest (ie, movement and PA vs rest and physical inactivity). Second, to check that the selected pictures were equivalent in term of credibility and agreeableness across categories (ie, movement vs rest). Pictures will be built to match for colour, brightness and visual complexity. To examine the generalisation of training effects,[72] in both the intervention and comparator group, only half of the pictures used in the assessment phase will be included on the training phase (the selected pictures will be counterbalanced across participants).

## Randomisation and blinding
The research assistants and the participants will be blinded to the allocation of the groups. At the end of the trial, the success of the participant blinding will be examined by asking the participants to guess in what group there were, including a percentage of certainty. Moreover, the success of research assistants blinding will be examined by asking each research assistant if they were able to detect the group (comparator vs intervention) when they conducted the data collection.

The randomisation will be generated on a computer and will be performed using permuted blocks (size=8). To ensure that the research team will be blinded to the randomisation, an independent coworker will carry out the randomisation. The patient's identification number will be used to determine the sequence of randomisation. Patients will be randomised in a 1:1 ratio between the intervention and active control condition. Unblinding is not planned during the trial as we do not see any reasons that would require either the patients or the researchers to know the group in which the patients were allocated. However, if requested by the patients, unblinding is permissible at the end of the trial.

## Outcomes
### Primary outcome

The primary outcome will be the sum of accelerometer-based time spent in light-intensity, moderate-intensity and vigorous-intensity PA over 1 week after the CBM intervention (in minutes per week). Following recommendations in patients,[73] a three-axis accelerometer (Actigraph GT3X+; Pensacola, USA) will be used to assess PA. Patients will be given the accelerometer and related indications during the first training session. They will be asked to wear the accelerometer for the full week and to return during the next appointment. They will be instructed on how to wear the device (ie, over the right hip, affixed to an elastic belt, preferably worn under their waistbands). Currently, the waist-mounted Actigraph is the most used device to objectively measure PA.[74] One-minute epochs will be used for data analyses and non-wear time will be defined as ≥59 consecutive minutes of zero counts. Daily data will be included if the wear time is ≥10 waking hours per day.[75] Data will be included if ≥4 days met the aforementioned conditions.[76] The time spent in light, moderate and vigorous PA over the week will be determined through previously validated cut points,[77] in bouts lasting at least 10 min. Then, in the week following the rehabilitation period, participants will be asked to wear the accelerometer for 1 week. The sum of times spent in light-intensity, moderate-intensity and vigorous-intensity PA during this period (in minutes per week) will be used as the primary outcome. Of note, because the duration of the rehabilitation period may strongly vary between patients, it is possible that some patients will be still in the hospital after 3 weeks, while other will leave the service sooner (eg, at 2 weeks). As such, to account for this feature and to allow comparisons between patients, the accelerometer will be scheduled to start on the Monday following their discharge from the rehabilitation unit, regardless the lengths of stay in the ward. Finally, participants will be asked to wear the accelerometer for 1 week at 1, 3, 6 and 12 months postintervention.

### Secondary outcomes

The secondary outcomes will be the changes in (1) automatic approach tendencies and self-reported motivation to be active, (2) physical health and (3) mental health. Sedentary behaviours and self-reported PA will also be examined. Table 2 provides an overview of all the outcomes measures and table 3 provides the schedule of assessment.

## Data analysis
### Primary analyses

Statistical analyses will be performed according to the intention-to-treat principle and will abide by the Consolidated Standards of Reporting Trials (CONSORT) guidelines. Analysis will be conducted in a blinded way. We will use mean, SD, median and range values to summarise the continuous data. The primary outcome (ie, the time spent in light-intensity, moderate-intensity and vigorous-intensity PA over 1 week after the CBM intervention) will be analysed using multiple linear regressions. Specifically, to test H1, we will test whether the patients' PA level during the week after the end of the intervention will be higher in the intervention group relative to the comparator group, after adjustment for covariates (ie, age, sex and indicators of the medical evaluation during the screening assessment). To test H2a and H2b, we will test whether patients' automatic approach tendencies towards PA will be higher and patients' automatic approach tendencies towards sedentary behaviours will be lower in the intervention group relative to the comparator group, after adjustment for covariates. Finally, to test H3, we will test whether patients' physical and mental health during the week after the end of the intervention will be higher in the intervention group relative to the comparator group, after adjustment for covariates. Moderator analyses (ie, for motivation to change, usual level of PA, personality, expectations for improvement) will be conducted.

### Secondary analyses

The aforementioned models will be tested at 1, 3, 6 and 12 months after the intervention. Moreover, to examine the effect of the intervention during the rehabilitation period, mixed effects models will be used. These models account for the nested structure of the data (ie, multiple observations within a single participant), thereby providing accurate parameter estimates with acceptable type I error rates.[78] Moreover, these models do not require an equal number of observations across participants, thereby allowing participants with missing observations to be included in the analyses without the need to impute those missing data. To formally examine the effect of the intervention on the evolution of PA within the rehabilitation period, models will include interaction terms between conditions (intervention group vs comparator group) and number of days within the rehabilitation programme (linear and quadratic). The number of days should be relatively equal between patients (about 21 days) but may differ to some extent (some patients can leave earlier or other later than 21 days). A statistically significant interaction will indicate that the rate of PA change throughout the rehabilitation programme would be different across the conditions. The quadratic effect of the number of days will be included to account for potential non-linear change of PA across the rehabilitation period. This will allow, for instance, to model the possibility that the effect of the intervention will take some sessions before becoming effective or that no additional effect could be hoped after a certain number of sessions. The continuous secondary outcomes will be treated in the similar way to the primary outcome. All analyses will be conducted using R software. Any deviation from the original statistical plan will be described and justified in the final trial report.

## Data security, management and monitoring

Project data will be handled with uttermost discretion and will be only accessible to authorised personnel who

**Table 2** Outcomes measures of the IMPACT trial and assessment time point

| Outcome | Assessment method | Measurement time points* |
|---|---|---|
| **Primary outcome** | | |
| PA | Accelerometer-based PA (Actigraph GT3X+) to measure the time spent in light-intensity, moderate-intensity and vigorous-intensity PA over 1 week. | During the rehabilitation, 1 week after, as well as 1, 3, 6 and 12 months after the intervention. |
| **Secondary outcomes** | | |
| **Automatic precursors of PA** | | |
| Approach tendencies | The Visual-Approach/Avoidance-by-the-Self Task.[71] A computerised reaction-time task assessing automatic approach tendencies towards PA and sedentary behaviours. | During the rehabilitation, 1 week after, as well as 1, 3, 6 and 12 months after the intervention. |
| **Physical health** | | |
| Weight | Weight (accuracy 0.1 kg) is assessed with participants clothed (lightweight clothing) | During the rehabilitation, 1 week after, as well as 1, 3, 6 and 12 months after the intervention. |
| Muscle strength | Grip strength measured with a handheld dynamometer.[84] | |
| Cardiorespiratory fitness | Maximal graded exercise test.[85] | |
| Perceived global physical health | Global physical health Patient-Reported Outcomes Measurement Information System (PROMIS) scale (one item: 'In general, how would you rate your physical health?'). | |
| **Mental health** | | |
| Perceived physical functioning, fatigue, self-efficacy towards PA | Physical and fatigue PROMIS scales and perceived capability from the Multi-process action control approach,[86] (six items. for example, 'To what extent are you able to carry out your everyday physical activities such as walking, climbing stairs, carrying groceries, or moving a chair?'; 'I have the physical ability to walk around the hospital'; 'In the past 7 days, how would you rate your fatigue on average?'). | During the rehabilitation, 1 week after, as well as 1, 3, 6 and 12 months after the intervention. |
| Perceived pain interference | Pain interference and pain intensity PROMIS scales (one item: 'In the past 7 days, how would you rate your pain on average?') | |
| Depression, anxiety, general life satisfaction | Anxiety, depression, general life satisfaction PROMIS scales (eight items. for example, 'In general, how would you rate your mental health, including your mood and your ability to think?'; 'In general, how would you rate your quality of life?'). | |
| Sleep | Sleep disturbance PROMIS scales (two items: 'In the past 7 days, my quality of sleep was…'; 'How satisfied/dissatisfied are you with your current sleep?'). | |
| Social role | Ability to participants in social roles and activities PROMIS scale (one item: 'In general, how would you rate your satisfaction with social activities and relationships?'). | |
| **Other PA-related measures** | | |

**Table 2** Continued

| Outcome | Assessment method | Measurement time points* |
|---|---|---|
| Self-reported behaviours | The International PA Questionnaire to measure the time spent in PA and in sedentary behaviours.[87] | During the rehabilitation, 1 week after, as well as 1, 3, 6 and 12 months after the intervention. |
| Sedentary behaviours | Accelerometer-based sedentary behaviours (Actigraph GT3X+) | |
| Attitudes | Instrumental (two items: useful, beneficial) and affective (two items: enjoyable, interesting) attitudes towards PA using a short self-reported questionnaire.[88 89] | |
| Self-reported motivation | Intention (one item: 'To what extent do you intend to do physical activities (such as walking in the hospital or in the park) during your rehabilitation?') and importance (one item: 'How important is it for you to engage in physical activity during your rehabilitation?'). | |

The main time point of analysis will be the week after the end of intervention.
PA, physical activity.

require the data to fulfil their duties within the scope of the research project. On the online CRFs and other specific documents, participants are only identified by a unique participant number. The online CRF will be created using REDCap.

Data recording: The dataset will be accompanied by a README file, which will describe the directory hierarchy and file naming convention. The directory will contain an INFO file describing the experimental protocol used in that experiment. This INFO file will also record any deviation from the protocol and other useful contextual information. This procedure should allow the data to be easily understood by other researchers and should support future reuse of the data. Metadata will be created to provide contextual information required to interpret data. This metadata file will be created in accordance with the data documentation initiative. In particular, the metadata file will include short unique identifier, the name of the author(s), the content, the date of creation, the locations, the reason why the data were generated, and how the data were created. The codebook will explicitly indicate the name, explanations and the modalities of the different variables measured in the experiment. In addition, it will include information on the study design and contain all information necessary for another analyst to use the data accurately.

Data anonymisation: Individual participant information collected during the study is considered confidential and disclosure to third parties is prohibited. Subject confidentiality will be ensured by using subject identification code numbers to correspond to treatment data in the computer files. Only a minority of personnel (ie, the principal investigator and chief medical officer) will have access to the data in a non-coded form.

Data storage: Participant data on a secure database in accordance with the General Data Protection Regulations (2018). Three copies of the data will be stored. First, original data will be stored on the principal investigator's computer, which will be backed up daily, and protected by a password. Additionally, data will be stored on a secure server hosted by the University of Geneva. Finally, data will be stored on an external device at a different location and be protected by a password. The original notebook will be stored in the principal investigator's laboratory. Local version of the data for statistical analysis will remain on a university computer, and be password protected. Each person who collected the data will have the responsibility to annotate their data within the metadata. Nevertheless, the principal investigator will have the responsibility to weekly check that the data is properly processed, documented and stored. All study data will be archived for a minimum of 10 years beyond the end of the randomised controlled trial.

Trial monitoring: The PI will organise a proper training of all involved study personnel to ensure that the study will be conducted according to the protocol. Research assistants should understand the detailed contents of the protocol before starting the data collection. For quality assurance the ethics committee may visit the research sites. Direct access to the source data and all project-related

**Table 3**  Schedule of assessment

**Week 1**

| Visit | Information (−1 day) | Screening (0) | First training session (+1) | Second training session (+2) | Third training session (+3) | Fourth training session (+4) |
|---|---|---|---|---|---|---|
| Oral and written patient information | + | | | | | |
| Informed written consent | | + | | | | |
| Inclusion exclusion criteria | | + | | | | |
| Additional baseline screening assessment | | + | | | | |
| Self-reported PA (usual week) | | + | | | | |
| Intervention | | | + | + | + | + |
| Motivation to be active | | | + | | + | + |
| Approach tendencies | | | + | | | |
| Physical health | | | + | | | |
| Mental health | | | + | | | |
| Accelerometer-based PA | | | Continuously across the week | | | |

**Week 2**

| Visit | First training session (+1) | Second training session (+2) | Third training session (+3) | Fourth training session (+4) |
|---|---|---|---|---|
| Intervention | + | + | + | + |
| Motivation to be active | + | + | | |
| Approach tendencies | + | | | |
| Physical health | + | | | |
| Mental health | + | | | |
| Accelerometer-based PA | Continuously across the week | | | |

**Week 3**

| Visit | First training session (+1) | Second training session (+2) | Third training session (+3) | Fourth training session (+4) |
|---|---|---|---|---|
| Intervention | + | + | + | + |
| Motivation to be active | + | + | | |
| Approach tendencies | + | | | |
| Physical health | + | | | |
| Mental health | + | | | |
| Accelerometer-based PA | Continuously across the week | | | |

**Postintervention**

| | 1 week | 1 month | 3 months | 6 months | 12 months |
|---|---|---|---|---|---|
| Motivation to be active | + | + | + | + | + |

Continued

**Table 3** Continued

| Week 1 | −1 day | 0 | +1 | +2 | +3 | +4 |
|---|---|---|---|---|---|---|
| Visit | Information | Screening | First training session | Second training session | Third training session | Fourth training session |
| Approach tendencies | + | + | + | + | + | |
| Physical health | + | + | + | + | + | |
| Mental health | + | + | + | + | + | |
| Self-reported and accelerometer-based PA (during 1 week) | + | + | + | + | + | |

PA, physical activity.

files and documents must be granted on such occasions. The principal investigator or any other competent authority may terminate the study prematurely according to the following circumstances: ethical concerns, insufficient participants recruitment, early evidence of harm or benefit of the experimental intervention through the interim analysis planned at 6 months after the start of the trial. Although no serious adverse event resulting from the intervention is expected, all potential adverse events will be documented within the eCRF.

### Patient and public involvement in the trial design
No patient or public was involved in the present study.

### Ethics and dissemination
The study was approved by the Ethics Committee of Geneva Canton, Switzerland (reference number: CCER2019-02257). All participants will give an informed consent to participate in the study.

Results will be published in relevant scientific journals and be disseminated in international conferences. Anonymity of the participants will be guaranteed when presenting the data at scientific meetings or publishing them in scientific journals. Individual participant information collected during the study is considered confidential and disclosure to third parties is prohibited.

Data sharing and reuse: Datasets and metadata from this trial will be deposited in ZENODO (a generic and free repository based at CERN, Geneva), and made public at the time of publication. Data in the repository will be stored in accordance with funder and university data policies. Particularly, original datasets, original software script and code, and original raw data will be deposited. However, as stressed above, personal data will be anonymised before diffusion.

### DISCUSSION
PA is associated with a wide range of health benefits,[1–7] but patients, similarly to the general population, remain largely physically inactive. Promoting PA to patients is thus urgently warranted, and healthcare professionals are uniquely placed to do so.[8] To date, interventions mainly rely on providing rational information to change patients' conscious goals and motivation to be active. Yet, these approaches are insufficient to substantially impact actual behaviours.[24] One explanation for this lack of effectiveness draws on recent observations suggesting that automatic reactions towards exercise-related stimuli are involved in the regulation of PA.[33 34 39 79 80] As such, developing interventions targeting both reflective (eg, motivation) and automatic (eg, approach tendencies) precursors of PA could be particularly effective. This protocol paper outlines the design of the IMPACT trial, the first placebo, triple-blinded, randomised controlled trial examining the effectiveness of a CBM intervention targeting automatic approach tendencies towards exercise-related stimuli on PA in patients in rehabilitation

programme remains. The IMPACT trial will focus on an accelerometer-based measure of PA as the primary outcome due to all the extensive benefits associated with being physically active. The secondary outcomes will allow examining other positive-side effects of the intervention on physical and mental health.

The IMPACT randomised controlled trial has several strengths. First, it is the first randomised controlled trial investigating the beneficial effect of an easy deliverable CBM intervention promoting PA among patients enrolled in a multidisciplinary rehabilitation programme. Second, this CBM intervention is anchored within the dual-process models of behaviour, arguing that automatic reactions towards PA represent additional targets for interventions. Accordingly, this trial will examine for the first time the efficacy of these new types of interventions, which directly targets the automatic precursors of PA behaviour. Third, we relied on an accelerometer-based measure of PA, which guarantee the validity and reliability of our primary outcome. Finally, in addition to PA behaviour, we will collect data on physical and mental health at multiple time points over 1 year. However, potential limitations should be noted. The first limitation is related to the fact that the trial is based on a single centre, which will limit the generalisation of the results to other centres. Second, because of the longitudinal design (ie, the main time point for the main analysis is assessed 4 weeks after the start of the intervention and additional time point for secondary analyses are assessed 1, 3, 6 and 12 months after the start of the intervention), we cannot exclude a selection bias due to attrition. Likewise, as participation in our study is voluntary, it may favour the selection of patients with a higher health status or the most motivated to engage in PA. These features are key limitations that may reduce our ability to generalise the results to other populations. Third, to reduce patients' burden, the measure of physical and mental health is based on a single or few items, which may reduce the reliability and validity of these secondary outcomes' measurement. Finally, the rehabilitation programme in the Division of General Medical Rehabilitation is a programme receiving patients that have been in acute care for different reasons such as serious infections, cancer, heart or lung failure or postsurgery follow-up treatments. Accordingly, the profiles of the patients included in the trial may strongly differ from one patient to another. Therefore, although patients' profile (eg, age, sex or features of the medical evaluation) will be adjusted in the model, the diversity of those profiles may still produce a level of variability likely to influence the effects of the intervention.

PA is a key factor to improve the management of patients' diseases. Helping patients to become more active is likely to promote their recovery, their physical and mental health, as well as to reduce the development of other comorbidities. Targeting automatic reactions towards PA, which may be negatively biased in patients, is particularly innovative. Furthermore, this low cost and easily deliverable intervention could be rapidly implemented on a large scale to help patients become more physically active. The findings from this study will provide evidence-based conclusions for future interventions promoting PA in patients.

**Author affiliations**
[1]Swiss Center for Affective Sciences, University of Geneva, Geneva, Switzerland
[2]Laboratory for the Study of Emotion Elicitation and Expression (E3Lab), Department of Psychology, University of Geneva, Geneva, Switzerland
[3]Division of Rheumatology, Department of Medicine, Geneva University Hospitals, Geneva, Switzerland
[4]Laboratory SENS, Department of Sport Science, University Grenoble Alpes, Saint-Martin-d'Heres, France
[5]Population Health Laboratory, Department of Medicine, University of Fribourg, Fribourg, Switzerland
[6]Department of Readaptation and Geriatrics, University of Geneva, Geneva, Switzerland
[7]Department of Psychology, Saarland University, Saarbrucken, Germany
[8]Department of Psychology and center for Urban Mental Health, University of Amsterdam, Amsterdam, The Netherlands
[9]School of Rehabilitation Sciences, Faculty of Health Sciences, University of Ottawa, Ottawa, Ontario, Canada
[10]Bruyère Research Institute, Ottawa, Ontario, Canada
[11]Quality of Care Unit, Geneva University Hospitals, Geneva, Switzerland
[12]Division of General Medical Rehabilitation, University Hospitals of Geneva, Geneva, Switzerland

**Contributors** BC: conceptualisation, writing-original draft; AF: conceptualisation, writing-review and editing; SM: Methodology-creation of the tasks, writing-review and editing; LF: writing-review and editing; SC: writing-review and editing; DS: writing-review and editing; MF: writing-review and editing; RWW: writing-review and editing; MPB: conceptualisation, writing-original draft; DSC: supervision, resources-provision on instrumentation, writing-review and editing; CL: supervision, resources-provision of study materials, writing-original draft.

**Funding** This work was supported by the Swiss National Science Foundation, grant number PZ00P1_180040. The purchase of protocol-related material (ie, 30 Polar watches and accelerometers) is provided by the non-operating fund of the Division of General Medical Rehabilitation, University Hospital of Geneva, Switzerland. The computer equipment (3 laptops) will be made available by Division of General Medical Rehabilitation, University Hospital of Geneva. MPB is supported by the Natural Sciences and Engineering Research Council of Canada (RGPIN-2021-03153) and by a grant from The Banting Research Foundation.

**Competing interests** None declared.

**Patient consent for publication** Consent obtained directly from patient(s)

**Provenance and peer review** Not commissioned; externally peer reviewed.

**ORCID iDs**
Boris Cheval http://orcid.org/0000-0002-6236-4673
Layan Fessler http://orcid.org/0000-0002-8435-5110
Matthieu P Boisgontier http://orcid.org/0000-0001-9376-3071
Delphine S Courvoisier http://orcid.org/0000-0002-1956-2607

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
