## [Reviewer comments · BMJ Open]

ARTICLE DETAILS

TITLE (PROVISIONAL)	A cognitive-bias modification intervention to improve physical activity in patients following a rehabilitation program: protocol for the randomized controlled IMPACT trial
AUTHORS	Cheval, Boris ; Finckh, Axel; Maltagliati, Silvio; Fessler, Layan; Cullati, Stéphane; Sander, David; Friese, Malte; Wiers, Reinout W.; Boisgontier, Matthieu P.; Courvoisier, Delphine; Luthy, C

VERSION 1 – REVIEW

REVIEWER	Pfeffer, Ines MSH Medical School Hamburg
REVIEW RETURNED	24-Jun-2021

GENERAL COMMENTS	Thank you very much for the opportunity to read this interesting study protocol. The manuscript describes a study protocol of a randomized controlled clinical trial within an inpatient multidisciplinary rehabilitation program. The study wants to generate evidence for the effectiveness a training program aiming at increasing patients' automatic approach tendencies toward PA. To reach this goal the authors plan to conduct a a single-center, placebo (sham controlled), double-blinded, phase 3 randomized controlled trial. Theoretical background Page 7, lines 3-5: The authors state: "To the best of our knowledge, however, only one study has been conducted to examine the effect of a brief CBM intervention on an exercise task in a sample of healthy young adults." I was wondering if the study of Antoniewicz & Brand (2016) was testing a quite similar intervention? Antoniewicz & Brand (2016). Learning to like exercise: Evaluative Conditioning Changes Automatic Evaluations of Exercising and Influences Subsequent Exercising Behavior. Journal of Sport and Exercise Psychology, 38, 138-148. Page 7, lines 58-60. "The authors hypothesize that that the CBM intervention will increase patients' self-reported motivation to be active (H2b)" Based on the theoretical background it is not clear for the reader how an intervention targeting automatic approach tendencies can influence a more reflective variable such as self-reported motivation. Aren't these different things? How can it happen that training of automatic approach tendencies can change reflective motivation? Please explain in the theoretical background.
--

Methods:

Page 8

Please also provide information about the average duration of hospitalization of the participants and also about the health issues/indications the patients show.

Page 9, lines 43-50

Please indicate the points of measurement included in the study and the statistical analyses on which the simple size estimation was based on.

The authors state: "We expect a loss to follow-up of 10 to 20% over one year." I think this is quite optimistic. Please provide a rational or a reference for this drop-out rate over one year.

Page 9, lines 55-58

"We expect that 1 patient out of 5 will not agree (for various reasons) to participate in the study". Please justify this expectation.

Page 10, line 26

"Training program of 15 sessions over 3 weeks (i.e., 5 sessions by week on average)". In the abstract, the authors are talking about 12 sessions. Please check and adapt accordingly.

Furthermore, how is training completion/compliance with the treatment controlled for?

Page 10, lines 32/33

Could you please also give examples of pictures showing 'landscape' (Figure 2; Appendix 2)

Page 11, lines 17ff

I agree, that pretesting the stimuli is very important. However, is this reflective approach used in the pretest valid to reflect the more automatic processes of PA behavior?

Page 12, line 19ff

In the appendices and the ethics application the authors indicate to use Polar system (see also pages 9/29, 12/29; 25/29) in order to assess PA and SB. What is actually used ActiGraph or Polar? And how valid and reliable are these data?

Furthermore, in order to be able to assess sitting, standing, light, moderate and vigorous PA with an accelerometer, the device should be placed at a significant position of the body. Therefore, please indicate where participants will wear the device (hip?), because with regard to this position data might not be valid for some of these behaviors.

For the secondary outcomes: Please indicate reliability and validity of the scales and measures.

Page 12/13

Please indicate how each hypothesis is statistically modelled in order to be tested.

Page 15/16

	Please also include limitations of the study and expected risk of bias. Page 22 In the screening assessment some constructs are measured (e.g., personality), without saying something about the respective hypotheses. Please indicate the respective Hypothesis and statistical analyses with regard to these measures. More details about the measures are needed. Number of items, item examples, answering format, reliability and validity of the scale, reference (e.g., self-reported motivation to change, Self-reported ability to implement daily-life PA). Page 23 Self-reported motivation was defined as a secondary outcome of the study. How is this variable measured? How is Perceived physical functioning measured? Page 24 The table only includes 4 training sessions per week. It is not clear how many training sessions are actually planned? 15 or 12? Flow-chart Again, 12 training sessions are indicated. What is the right number of training sessions? Furthermore, the authors talk about daily assessments during the 3-week rehabilitation program. I could not find any variable that is assessed on a daily basis? Which assessments do you mean? Page 5/29, line 46 The use of healthcare system is mentioned as a secondary outcome for the first time. How is this measured and why? Page 13/29, lines 23f The authors state that statistical analysis will be conducted in a blinded way. Is the RCT triple-blinded instead of double-blinded then? In General: The authors could also indicate the expected/estimated 'risk of bias' of this study based on the following publication: Sterne JAC, Savović J, Page MJ, Elbers RG, Blencowe NS, Boutron I, Cates CJ, Cheng H-Y, Corbett MS, Eldridge SM, Hernán MA, Hopewell S, Hróbjartsson A, Junqueira DR, Jüni P, Kirkham JJ, Lasserson T, Li T, McAleenan A, Reeves BC, Shepperd S, Shrier I, Stewart LA, Tilling K, White IR, Whiting PF, Higgins JPT. RoB 2: a revised tool for assessing risk of bias in randomised trials. BMJ 2019; 366: l4898.
--	---

REVIEWER	Hannan, Thomas Griffith University
REVIEW RETURNED	30-Jun-2021

GENERAL COMMENTS	This looks to be a very promising study with great potential and impact. The protocol was very well written, but I just have a couple of questions for the authors to consider. (1) Can the authors provide a little more detail about the participants who will be recruited. That is, what is the primary reason why they are receiving rehabilitation? This would provide clarity about those participating in the study and also help to identify potential barriers to PA engagement which may be specific to that population. (2) Page 10, line 19 – Is it possible then that any change in self-reported motivation could be the result of the tracking device and personalized feedback, and not the result of the CBM intervention? See also my comment below (4). (3) Page 10, line 42 – Can the authors provide a rationale for why a 90%/10% split wasn't used for the intervention group (e.g., 90% approach trials, 10% avoid trials). This would help keep participants blinded to their condition when they are asked at the end of the intervention (see page 11, line 52 of protocol). (4) Can the authors provide insight into the type of rehabilitation that patients will receive? E.g., does the rehabilitation centre focus on increasing PA engagement through certain motivational interviewing or CBT techniques? It is indicated that patients may leave earlier or later than the expected 21-day rehabilitation, so it would be important to identify whether any change in (a) approach bias, (b) explicit motivation, and (c) behaviour, is the result of the CBM and not the rehabilitation program. For instance, Chevance et al., (2017; http://dx.doi.org/10.1037/rep0000137) observed changes in social cognition constructs and implicit attitudes towards PA following a rehabilitation program.
---

VERSION 1 – AUTHOR RESPONSE

Reviewer 1:

Dr. Ines Pfeffer, MSH Medical School Hamburg

Response: We thank the reviewer for the positive assessment of our work, and we appreciate the provision of comments that helped us to clarify and significantly improve the quality of the manuscript.

Theoretical background

1. Page 7, lines 3-5: The authors state: "To the best of our knowledge, however, only one study has been conducted to examine the effect of a brief CBM intervention on an exercise task in a sample of healthy young adults." I was wondering if the study of Antoniewicz & Brand (2016) was testing a quite similar intervention? Antoniewicz & Brand (2016). Learning to like exercise: Evaluative Conditioning Changes Automatic Evaluations of Exercising and Influences Subsequent Exercising Behavior. *Journal of Sport and Exercise Psychology*, 38, 138-148.

Response: We agree with the reviewer that, although evaluative conditioning cannot be directly comparable with approach-avoidance training task, this literature needs to be cited as it is relevant prior works. Additional references have been thus added including the aforementioned paper.

Page 7: "To the best of our knowledge, however, only a handful set of studies have been conducted to target automatic processes toward physical activity.⁶⁰⁻⁶³ Crucially, only one study has been conducted to examine the effect of a brief CBM intervention targeting approach-avoidance tendencies on an exercise task in a sample of healthy young adults.⁶³"

2. Page 7, lines 58-60. "The authors hypothesize that that the CBM intervention will increase patients' self-reported motivation to be active (H2b)" Based on the theoretical background it is not clear for the reader how an intervention targeting automatic approach tendencies can influence a more reflective variable such as self-reported motivation. Aren't these different things? How can it happen that training of automatic approach tendencies can change reflective motivation? Please explain in the theoretical background.

Response: The reviewer is right. Based on the available literature, it seems unreasonable to make such a priori hypothesis. The available evidence supports the hypothesis that CBM intervention may impact the automatic precursors of physical activity, but no robust data have examined its effect on the controlled precursors. This hypothesis has been deleted.

Methods:

3. Page 8. Please also provide information about the average duration of hospitalization of the participants and also about the health issues/indications the patients show.

Response: This information has been added. Thank you.

Page 10: "The average duration of participants hospitalization in the ward 3DK is about three weeks. As such, though this duration can vary between patients (i.e., some patients only stay a few days), this duration allows for the implementation of the whole intervention (i.e., 12-session training program performed over three weeks). Of note, participants who will not complete all the training sessions will still be included in the analysis. Sensibility analyses will be conducted to examine whether the number of completed sessions influence the effects of the intervention."

4. Page 9, lines 43-50. Please indicate the points of measurement included in the study and the statistical analyses on which the sample size estimation was based on. The authors state: "We expect a loss to follow-up of 10 to 20% over one year." I think this is quite optimistic. Please provide a rational or a reference for this drop-out rate over one year.

Response: We agree with the reviewer that the sample size calculation should be described with more details. Moreover, we now provide a more realistic attrition level.

Pages 9-10: "Sample size

For power calculation, our intervention implements a between-subject design and random-effects statistical models (i.e., t-tests). Considering a conservative medium effect size (Cohen's $d = 0.45$) a

sample size calculation indicates that a minimum of 220 patients (110 per arm) would be needed to demonstrate efficacy of the intervention on the device-based PA during the week following the intervention, with a probability of committing a type I error < 5% and a probability of committing a type II error < 10%. We expect a loss to follow-up of 10 to 20% at one week after the intervention, and a loss of 30 to 40% over one year. Thus, a minimum 308 patients will be recruited.”

5. Page 9, lines 55-58 “We expect that 1 patient out of 5 will not agree (for various reasons) to participate in the study”. Please justify this expectation.

Response: We agree with the reviewer that a justification is lacking. Moreover, based on previous experiences of the chief medical officer and a first presentation of the study to the patients of the unit, this number has been revised upwards. Accordingly, the manuscript has been amended as follow:

Page 10: “Feasibility

The ward 3DK of the Division of General Medical Rehabilitation has 24 beds and treats on average 40 patients per month. Based on the chief medical officer’s experiences and a first presentation of the study to the patients treated in this unit, we expect that 3 patients out of 5 will not agree (for various reasons) to participate in the study, thereby leading to a total of about 24 participants recruited per month. Consequently, we should be able to collect the target sample size in approximately 12-14 months.”

6. Page 10, line 26. “Training program of 15 sessions over 3 weeks (i.e., 5 sessions by week on average)”. In the abstract, the authors are talking about 12 sessions. Please check and adapt accordingly.

Response: We initially planned to have 15 sessions over 3 weeks, but to improve the trial feasibility, without compromising trial effectiveness (see Eberl et al. Alcohol. Clin. Exp. Res. 2014), we have decided to plan 12 sessions over the 3 weeks.

7. Furthermore, how is training completion/compliance with the treatment controlled for?

Response: We thank the reviewer for this comment. This information was indeed lacking. Participants adherence to the training program as well as all the other measures are rigorously documented in the electronic Case Report Form (eCRF) powered by REDCap.

Pages 10-11: “Patients adherence to the IMPACT trial

Patients adherence to the training program (i.e., if the planned training session is completed or not, and why in case of no completion) and to the other measures are documented in an electronic Case Report Form (eCRF) powered by REDCap.⁶⁷ To promote patient retention and complete follow-up (i.e., one, three, six and 12 months after the end of the intervention), participants will be contacted by phone by a research assistant two weeks before the follow-up measurement. If they do not answer, they will receive up to two additional phone calls this week. If they do not answer, this procedure will be repeated the following week. If they still do not answer, this time of measurement will be considered as missing. Patients with missing data at a given wave, will be contacted for the following waves through the above-mentioned procedure.”

8. Page 10, lines 32/33 Could you please also give examples of pictures showing 'landscape' (Figure 2; Appendix 2).

Response: We are not sure to understand this comment as the Figure 2 already included both pictures in the portrait format and in the landscape format.

9. Page 11, lines 17ff. I agree, that pretesting the stimuli is very important. However, is this reflective approach used in the pretest valid to reflect the more automatic processes of PA behavior?

Response: We thank the reviewer for this comment. The objective of the pre-test was to identify to what extent the pictures adequately reflected “movement and physically active behaviors” and “rest and physically inactive behaviors”. In other words, we were not interested in examining the automatic evaluation of these pictures, but we simply aimed to ensure that the pictures were perceived to reflect the concept targeted (i.e., movement vs. rest) as indicated by pretest participants’ self-reports. That is, the objective is to ensure that the pictures selected were indeed related to the concept of physical activity (vs. rest). Likewise, the measure of the credibility and the agreeableness of each picture has been carried out to ensure that the pictures of “movement and physically active behaviors” are equivalent to those of “rest and physically inactive behaviors” on these two dimensions, as a difference on these dimensions may potentially act as a confounding factor. A sentence has been added to highlight the objective of the pilot study.

Page 12: “Stimuli:

The aim of this pilot study was twofold. First, to ensure that the selected pictures reflect the concept of interest (i.e., movement and physical activity vs. rest and physical inactivity). Second, to check that the selected pictures were equivalent in terms of credibility and agreeableness across categories (i.e., movement vs. rest).”

10. Page 12, line 19. In the appendices and the ethics application the authors indicate to use Polar system (see also pages 9/29, 12/29; 25/29) in order to assess PA and SB. What is actually used ActiGraph or Polar? And how valid and reliable are these data?

Furthermore, in order to be able to assess sitting, standing, light, moderate and vigorous PA with an accelerometer, the device should be placed at a significant position of the body. Therefore, please indicate where participants will wear the device (hip?), because with regard to this position data might not be valid for some of these behaviors.

For the secondary outcomes: Please indicate reliability and validity of the scales and measures.

Response: We thank the reviewer for this comment. Initially, we planned to use the Polar in order to measure physical activity. This decision was made as we wanted to give to the patients a device providing them feedback about their physically active and inactive behaviors. However, we have tested this device on ourselves and on some patients, and we were not convinced about the validity of the data. This feeling was reinforced by a recent publication (Degroote et al. JMIR Mhealth UHealth. 2018) showing a systematic bias for the smartwatches-Polar M600 in measurement of moderate-to-vigorous physical activity. Accordingly, we have decided to use the Actigraph, an accelerometer to measure physical activity, which has been the most commonly used device to objectively assess physical activity in the clinical setting and in everyday life (Bassett et al. Med. Sci. Sports Exerc. 2015). The Polar device was kept but only as a tool to provide feedback to the patients regarding their physical activity behaviors. As suggested, we have added information on how to wear the accelerometer.

Pages 13-14: “Primary outcomes

The primary outcome will be the accelerometer-based time spent in PA. Following recommendations in patients, a three-axis accelerometer (Actigraph GT3X+; Pensacola, USA) will be used to assess PA. Patients will be given the accelerometer and related indications during the first training session. They will be asked to wear the accelerometer for the full week and to return during the next appointment. They will be instructed on how to wear the device (i.e., over the right hip, affixed to an elastic belt, preferably worn

under their waistbands). Currently, the waist-mounted Actigraph is the most used device to objectively measure physical activity.⁷⁰ One-minute epochs will be used for data analyses and non-wear time will be defined as ≥ 59 consecutive minutes of zero counts.

11. Page 12/13. Please indicate how each hypothesis is statistically modelled in order to be tested.

Response: As suggested, information on how each hypothesis will be tested has been added. Moreover, we now clearly separate the primary analyses (i.e., focusing on the outcomes during the week after the intervention) from the secondary analyses (i.e., focusing on the outcomes at one, three, six and 12 months after the end of the intervention, as well as during the rehabilitation period). Thank you.

Pages 14-15: "Data analysis

Primary analyses

Statistical analyses will be performed according to the intention-to-treat (ITT) principle and will abide by the Consolidated Standards of Reporting Trials (CONSORT) guidelines. Analysis will be conducted in a blinded way. We will use mean, standard deviation (SD), median, and range values to summarize the continuous data. The primary outcome will be analyzed using multiple linear regressions. Specifically, to test H1, we will test whether the patients' PA level during the week after the end of the intervention will be higher in the intervention group relative to the comparator group, after adjustment for covariates (i.e., age, sex, and indicators of the medical evaluation during the screening assessment). To test H2a and H2b, we will test whether patients' automatic approach tendencies toward PA will be higher and patients' automatic approach tendencies toward sedentary behaviors will be lower in the intervention group relative to the comparator group, after adjustment for covariates. Finally, to test H3, we will test whether patients' physical and mental health during the week after the end of the intervention will be higher in the intervention group relative to the comparator group, after adjustment for covariates. Moderator analyses (i.e., for motivation to change, usual level of PA, personality, expectations for improvement) will be conducted.

Secondary analyses

The aforementioned models will be tested one, three, six, and 12 months after the intervention. Moreover, to examine the effect of the intervention during the rehabilitation period, mixed effects models will be used. These models account for the nested structure of the data (i.e., multiple observations within a single participant), thereby providing accurate parameter estimates with acceptable Type I error rates.⁷⁴ Moreover, these models do not require an equal number of observations across participants, thereby allowing participants with missing observations to be included in the analyses without the need to impute those missing data. To formally examine the effect of the intervention on the evolution of PA within the rehabilitation period, models will include interaction terms between conditions (intervention group vs. comparator group) and number of days within the rehabilitation program (linear and quadratic)."

12. Page 15/16. Please also include limitations of the study and expected risk of bias.

Response: As suggested, the methodological limitations of the study have been added

Pages 18-19: "Strengths and limitations:

“However, potential limitations should be noted. The first limitation is related to the fact that the trial is based on a single center, which will limit the generalization of the results to other centers. Second, because of the longitudinal design (i.e., the main time point for the main analysis is assessed four weeks after the start of the intervention and additional time point for secondary analyses are assessed one, three, six, and 12 months after the start of the intervention), we cannot exclude a selection bias due to attrition. Likewise, as participation in our study is voluntary, it may favor the selection of patients with a higher health status or the most motivated to engage in PA. These features are key limitations that may reduce our ability to generalize the results to other populations. Third, to reduce patients’ burden, the measure of physical and mental health is based on a single or few items, which may reduce the reliability and validity of these secondary outcomes’ measurement. Finally, the rehabilitation program in the Division of General Medical Rehabilitation is a program receiving patients that have been in acute care for different reasons such as serious infections, cancer, heart or lung failure, or post-surgery follow-up treatments. Accordingly, the profiles of the patients included in the trial may strongly differ from one patient to another. Therefore, although patients’ profile (e.g., age, sex, or features of the medical evaluation) will be adjusted in the model, the diversity of those profiles may still produce a level of variability likely to influence the effects of the intervention.”

13. Page 22. In the screening assessment some constructs are measured (e.g., personality), without saying something about the respective hypotheses. Please indicate the respective Hypothesis and statistical analyses with regard to these measures.

More details about the measures are needed. Number of items, item examples, answering format, reliability and validity of the scale, reference (e.g., self-reported motivation to change, Self-reported ability to implement daily-life PA).

Response: All the constructs included on the screening assessment will be used as a moderator or as a covariate in the models. For example, we will explore whether personality moderates the effects of the intervention (see Page 13 at the end of the “primary analyses” section). Moreover, as suggested, more details and/or a reference have been provided on the measures (see Tables 1 and 2).

Page 25: “Table 1. Overview of the baseline screening measures

Measures Assessment method

Inclusion criteria

Patients treated in ward 3DK of the Division of General Medical Rehabilitation

During the first meeting with the research assistant.

≥ 18 years of age

Can comply with study protocol

Able to provide a written consent

Exclusion criterion

Contraindication to PA in the view of the health status During the first meeting with the research assistant.

Additional baseline screening assessment

Medical evaluation (questionnaires and objective tests) Patients’ diseases and treatment characteristics (medical burden, comorbidity, body mass index, mobility test, functional independence, health-related quality of life).

Sociodemographic characteristics Questionnaires (age, sex, height, weight).

Usual level of PA Saltin-Grimby PA Level Scale (SGPALS).77

Personality Ten-Item Personality Inventory (TIPI).78

Expectations for improvement

A questionnaire measuring patients' thoughts about the effects of the intervention (three items: "to what extent do you think that your physical activity behaviors will improve as a result of training on the computerized task?"; "to what extent do you think that your mental health will improve as a result of training on the computerized task?"; "to what extent do you think that your physical health will improve as a result of training on the computerized task?").66

Self-reported motivation to change Questionnaire measuring patients' motivation to change their condition (two items: "how motivated are you to change your health condition?"; "to what extent do you really want to change your health condition?"), to avoid a new treatment (two items: "how motivated are you to avoid a new treatment because your health condition?"; "to what extent do you really want to avoid taking a new medication because of your health condition?", and to engage in more PA in the future (two items: "I intend to carry out more physical activity in the next future"; "I am determined to carry out more physical activity in the next future").66

Self-reported ability to implement daily-life PA Questionnaire measuring patients' self-reported ability to adopt regular PA in their daily life. Self-reported function in instrumental activities of daily life (IADL; seven items), in activities of daily living (ADL; seven items), and in mobility (three items).79

Pages 26-27: "Table 2. Outcomes measures of the IMPACT trial and assessment time point"

Outcome Assessment method Measurement timepoints*

Primary outcomes

PA Accelerometer-based PA (Actigraph GT3X+) to measure the time spent in light, moderate, and vigorous PA. During the rehabilitation, one week as well as one, three, six and 12 months after the intervention.

Secondary outcomes

Automatic precursors of PA

Approach tendencies The Visual-Approach/Avoidance-by-the-Self Task (VAAST).67 A computerized reaction-time task assessing automatic approach tendencies toward PA and sedentary behaviors. During the rehabilitation, one week as well as one, three, six and 12 months after the intervention.

Physical Health

Weight Weight (accuracy 0.1 kg) is assessed with participants clothed (lightweight clothing) During the rehabilitation, one week as well as one, three, six and 12 months after the intervention.

Muscle strength Grip strength measured with a handheld dynamometer.80

Cardiorespiratory fitness Maximal graded exercise test.81

Perceived global physical health Global physical health Patient-Reported Outcomes Measurement Information System (PROMIS) scale (one item: "in general, how would you rate your physical health?").

Mental health

Perceived physical functioning, fatigue, self-efficacy toward PA Physical and fatigue PROMIS scales and perceived capability from the Multi-process action control approach,82 (six items. e.g., "To what extent are you able to carry out your everyday physical activities such as walking, climbing stairs, carrying groceries, or moving a chair?"; "I have the physical ability to walk around the hospital"; "In the past 7 days, how would you rate your fatigue on average?").

During the rehabilitation, one week as well as one, three, six and 12 months after the intervention.
Perceived pain interference Pain interference and pain intensity PROMIS scales (one item: "In the past 7 days, how would you rate your pain on average?")

Depression, anxiety, general life satisfaction Anxiety, depression, general life satisfaction PROMIS scales (eight items. e.g., "In general, how would you rate your mental health, including your mood and your ability to think?"; "In general, how would you rate your quality of life?").

Sleep Sleep disturbance PROMIS scales (two items: "In the past 7 days, my quality of sleep was..."; "How satisfied/dissatisfied are you with your current sleep?").

Social role Ability to participate in social roles and activities PROMIS scale (one item: "In general, how would you rate your satisfaction with social activities and relationships?")

Other PA-related measures

Self-reported behaviors The International PA Questionnaire to measure the time spent in PA and in sedentary behaviors.⁸³

During the rehabilitation, one week as well as one, three, six and 12 months after the intervention.

Sedentary behaviors Accelerometer-based sedentary behaviors (Actigraph GT3X+)

Attitudes Instrumental (two items: useful, beneficial) and affective (two items: enjoyable, interesting) attitudes toward PA using a short self-reported questionnaire.^{84 85}

Self-reported motivation Intention (one item: "To what extent do you intend to do physical activities (such as walking in the hospital or in the park) during your rehabilitation?") and importance (one item: "How important is it for you to engage in physical activity during your rehabilitation?")

Note. The main timepoint of analysis will be the week after the end of intervention

14. Page 23. Self-reported motivation was defined as a secondary outcome of the study. How is this variable measured? How is Perceived physical functioning measured?

Response: This information has been added. Please see the updated table 2 provided in the comment #13.

15. Page 24. The table only includes 4 training sessions per week. It is not clear how many training sessions are actually planned? 15 or 12?

Response: The trial will include four training sessions per week. Please see our response to comment #6.

Flow-chart

16. Again, 12 training sessions are indicated. What is the right number of training sessions? Furthermore, the authors talk about daily assessments during the 3-week rehabilitation program. I could not find any variable that is assessed on a daily basis? Which assessments do you mean?

Response: The correct number of training sessions is twelve. Please see our response to comment #6. For the daily basis, we referred to our primary outcome (i.e., physical activity) that will be continuously assessed during the rehabilitation period with an accelerometer. A note has been added.

Page 30: "Figure 1. Flow chart

Note. The daily assessment refers to the measure of PA behaviors that will be continuously assessed during the rehabilitation period. The secondary outcomes will be assessed on a weekly basis."

17. Page 5/29, line 46. The use of healthcare system is mentioned as a secondary outcome for the first time. How is this measured and why?

Response: Thank you for this comment. The protocol has been updated after a discussion with the healthcare professionals involved in the RCT. In particular, we considered that we were not in a position to accurately assess this outcome because the number of days hospitalization is highly variable depending on patients' health status. Any references to the use of healthcare system have been thus deleted in the main manuscript. Thank you.

18. Page 13/29, lines 23f. The authors state that statistical analysis will be conducted in a blinded way. Is the RCT triple-blinded instead of double-blinded then?

Response: The reviewer is right. Double-blinded has been replaced by triple-blinded across the manuscript. Thank you.

19. In General: The authors could also indicate the expected/estimated 'risk of bias' of this study based on the following publication: Sterne JAC, Savović J, Page MJ, Elbers RG, Blencowe NS, Boutron I, Cates CJ, Cheng H-Y, Corbett MS, Eldridge SM, Hernán MA, Hopewell S, Hróbjartsson A, Junqueira DR, Jüni P, Kirkham JJ, Lasserson T, Li T, McAleenan A, Reeves BC, Shepperd S, Shrier I, Stewart LA, Tilling K, White IR, Whiting PF, Higgins JPT. RoB 2: a revised tool for assessing risk of bias in randomised trials. *BMJ* 2019; 366: l4898.

Response: We thank the reviewer for this very relevant paper. The limitations (i.e., risk of bias) of the current study have been added. Please see our response to comment #12.

Reviewer: 2

Dr. Thomas Hannan, Griffith University

This looks to be a very promising study with great potential and impact. The protocol was very well written, but I just have a couple of questions for the authors to consider.

Response: We thank the reviewer for the positive assessment of our work, and we appreciate the provision of comments that helped us to clarify and significantly improve the quality of the manuscript.

1. Can the authors provide a little more detail about the participants who will be recruited. That is, what is the primary reason why they are receiving rehabilitation? This would provide clarity about those participating in the study and also help to identify potential barriers to PA engagement which may be specific to that population.

Response: We thank the reviewer for this comment. More details have been added regarding why patients are receiving rehabilitation in the ward 3DK. Moreover, we now discuss in the limitation section that the fact that patients were admitted in the unit for multiple reasons lead to the potential inclusion of patients with very different profiles. Although this diversity may be a strength, it is also accompanied by a potentially high level of variability between individuals that may bias the effects of the intervention.

Page 8: "Study design

The ward 3DK admits and manages patients for treatments or diagnostics evaluations, especially after being in acute care for several reasons, such as serious infections, cancer, heart failure, or post-surgery follow-up treatments. This ward offers multidisciplinary treatment in rehabilitation (e.g., physiotherapists, occupational therapists, nutritionists) and does not focus on improving PA engagement. In other words, within the usual care, there is not any content specifically devoted to improve patients' PA level."

Page 18: "Strengths and limitations

Finally, the rehabilitation program in the Division of General Medical Rehabilitation is a program receiving patients that have been in acute care for different reasons such as serious infections, cancer, heart or lung failure, or post-surgery follow-up treatments. Accordingly, the profiles of the patients included in the trial may strongly differ from one patient to another. Therefore, although patients' profile (e.g., age, sex, or features of the medical evaluation) will be adjusted in the model, the diversity of those profiles may still produce a level of variability likely to influence the effects of the intervention."

2. Page 10, line 19 – Is it possible then that any change in self-reported motivation could be the result of the tracking device and personalized feedback, and not the result of the CBM intervention? See also my comment below (4).

Response: The reviewer is right. Moreover, based on the available literature, it seems unreasonable to make such a priori hypothesis on the effect of the CMB intervention on the controlled processes. This hypothesis has thus been deleted.

3. Page 10, line 42 – Can the authors provide a rationale for why a 90%/10% split wasn't used for the intervention group (e.g., 90% approach trials, 10% avoid trials). This would help keep participants blinded to their condition when they are asked at the end of the intervention (see page 11, line 52 of protocol).

Response: We thank the reviewer for this thoughtful comment. We agree that using a 90/10 split seems wiser. The manuscript has been this updated accordingly.

Page 11: "Intervention group:

Participants in the intervention group will receive training in which 90% of pictures depicting PA will be presented in the approach format (and 10% in the avoidance format), and 90% pictures depicting sedentary behaviors will be presented in the avoidance format (and 10% in the approach format). This 90/10 split aims to increase the patients blinding to the condition in which they will be assigned."

Page 30: "Figure 2.

In the intervention condition, 90% of the pictures depicting physical activity are presented in the approach format (10% in avoidance format), and 90% of the pictures depicting sedentary behaviors are presented in the avoidance format (10% in approach format). In the control condition, the pictures depicting physical activity and sedentary behaviors are equally distributed across formats (i.e., 50%-50%)."

4. Can the authors provide insight into the type of rehabilitation that patients will receive? E.g., does the rehabilitation centre focus on increasing PA engagement through certain motivational interviewing or CBT techniques? It is indicated that patients may leave earlier or later than the expected 21-day rehabilitation, so it would be important to identify whether any change in (a) approach bias, (b) explicit motivation, and (c) behaviour, is the result of the CBM and not the rehabilitation program. For instance, Chevance et al., (2017; <http://dx.doi.org/10.1037/rep0000137>) observed changes in social cognition constructs and implicit attitudes towards PA following a rehabilitation program.

Response: We thank the reviewer for raising this point. We now explain that the rehabilitation is a general rehabilitation unit admitting and managing patients with various profiles. Hypothesis relative the effects of the CBM intervention on reflective processes (e.g., intention) has been removed. Similarly, the number of days in rehabilitation was no longer used as a targeted outcome as we considered that we were not in a position to accurately assess this outcome as the number of days hospitalization is highly variable depending on patients' health status.

Regarding the potential changed associated with the rehabilitation program per se and not the CBM, we fully agree that main effects of the rehabilitation program on the various outcomes should be observed (e.g., explicit motivation, but also physical activity behaviors and automatic processes will be likely to change across the rehabilitation). However, it is important to note that because patients will be randomized between the control condition and the intervention condition, it is unlikely that a difference observed between these conditions will be due to the rehabilitation program (i.e., all the participants will follow exactly the same rehabilitation program, the only difference being of if the patients will complete a re-training task or a placebo). In other words, all the differences observed between the groups (i.e., control vs. intervention) will be attributable to the CBM (i.e., the only manipulated variable that allow to distinguish between the group).

VERSION 2 – REVIEW

REVIEWER	Pfeffer, Ines MSH Medical School Hamburg
REVIEW RETURNED	23-Aug-2021

GENERAL COMMENTS	I thank the authors for the detailed answers on my comments and think that my concerns were sufficiently processed. Thank you very much!
--